# High Frequency of Concomitant Bladder, Bowel, and Sexual Symptoms in Huntington’s Disease: A Self-Reported Questionnaire Study

**DOI:** 10.3390/jpm11080714

**Published:** 2021-07-25

**Authors:** Brenda G. Vicars, Alice B. Liu, Sarah Holt, Suman Jayadev, Thomas Bird, Claire C. Yang

**Affiliations:** 1Department of Urology, University of Washington, Seattle, WA 98195, USA; bvicars@uw.edu (B.G.V.); abliu@uw.edu (A.B.L.); sholt@uw.edu (S.H.); cyang@uw.edu (C.C.Y.); 2Department of Neurology, University of Washington, Seattle, WA 98195, USA; sumie@uw.edu; 3Department of Medicine, Medical Genetics, University of Washington, Seattle, WA 98195, USA; 4VA Puget Sound Health Care System, Geriatric Research Educational and Clinical Center, Seattle, WA 98108, USA

**Keywords:** Huntington’s disease, bladder, bowel, sexual function, questionnaire

## Abstract

Huntington’s disease (HD) can be associated with pathologic involvement beyond the striatum including the autonomic nervous system. Bladder, bowel, and sexual dysfunction have been reported independently in HD, but little is known about their concomitant occurrence. To document this concomitant phenomena, forty-eight subjects (54% male, ages 28–74 years, CAG repeat 38–61) with manifest/symptomatic HD completed detailed questionnaires regarding bladder, bowel, and sexual function. In total, 45 subjects (93.8%) reported symptoms in at least one organ system (bladder, bowel, or sexual), 13 (27.1%) reported symptoms in two systems, and 19 (39.6%) reported concomitant symptoms in all three systems. Urinary problems were most frequent in 42 subjects (87.5%) followed by lower bowel (60.4%) and sexual dysfunction (56.2%). Participants reporting concomitant symptoms were more likely to have longer duration of disease and lower Total Functional Capacity (TFC) scores. This study documents the high frequency of bladder, bowel, and sexual dysfunction in HD and the common occurrence of concomitance of these pelvic organ problems.

## 1. Introduction

Huntington’s disease (HD) is an autosomal dominant neurodegenerative disease caused by an increased number of trinucleotide repeats in the coding region of the Huntington gene. The mean age of disease onset is 40 years with the mean duration of disease of 21.4 years [1]. The clinical features of HD are commonly described as a triad of motor, cognitive, and psychological symptoms, which are attributed to neuronal loss in the striatum and other areas of the brain. However, sparse attention has been paid to the association of HD with development of pelvic organ dysfunction, encompassing bladder, bowel, and sexual organs or the presence of concomitant symptomatology [2,3,4,5,6,7]. Current texts of HD rarely mention the presence of bladder and bowel dysfunction in HD, and the reporting of sexual function is presented predominantly in the context of disordered psychiatric manifestations [8,9,10,11]. In many conditions of neurologic disease and injury, bladder, bowel, and sexual function are known to be simultaneously compromised due to the proximity of the central neuroanatomical pathways mediating pelvic organ function [12,13]. Nerve damage in HD results in autonomic as well as somatic dysfunction, which can impact the coordination of nerves and muscles controlling bladder, bowel, and sexual function [14,15,16].

In our clinical experience at a Huntington’s disease specialty clinic, we have consistently noted adverse patient reports of symptoms such as urinary frequency, urgency, incontinence (both urinary and fecal), constipation, chronic diarrhea, and sexual dysfunction among individuals with HD, often in combination. This is in contrast to the longstanding recognition of concomitant pelvic organ dysfunction in other neurological diseases such as Parkinson’s and multiple sclerosis [17,18].

In this study, we aimed to further understand the frequency of concomitant bladder, bowel, and sexual dysfunction in persons with HD through patient-reported questionnaires.

## 2. Methods

After institutional review board (IRB) approval of the study protocol (approval number STUDY00006315), men and women with symptomatic HD were recruited from the Huntington’s Disease Society of America (HDSA) Center of Excellence at the University of Washington, during standard clinical visits. Informed consent was obtained prior to enrollment. Persons with a history of pelvic cancer treatment or pelvic surgery were excluded, as well as those with any other concurrent neurological disorder.

At study entry, CAG repeats, age of disease onset, and length of symptomatic disease were recorded along with a current medication list and demographics. All subjects were scored during their clinical visit using the Total Functional Capacity (TFC) scale to measure disease stage. The TFC is a standardized rating scale component of the Unified Huntington’s Disease Rating Scale (UHDRS) that uses 5 different categories—capacity to work in occupation, manage finances, perform household chores, perform self-care tasks, and level of care needs—to measure disease function in persons with HD. Refs. [19,20] A higher total score represents higher function. A functional score of 13 indicates full capacity in all five categories, whereas a score of 0 indicates total incapacity. These scores are used to categorize 5 stages of disease. Stage 1 (11–13 points) and stage 2 (7–10 points) represent the early stages of disease. Stage 3 (3–6 points) represents mid-stage, and stage 4 (1–2 points) and stage 5 (0 points) represent late-stage disease.

All study participants were asked to complete a series of detailed questionnaires related to bladder, bowel, and sexual function (available on request). If needed, a family member or a clinic staff member assisted in recording participant responses.

Urinary symptoms were assessed using the Lower Urinary Tract Symptoms (LUTS) Tool, a comprehensive, validated instrument of 22 questions on urinary symptoms, including a corresponding bother scale for each of those questions [21]. The recall period for the questions was one week. The frequency of each symptom was captured on a 5-point Likert scale (“never”, “rarely”, “sometimes”, “often”, and “almost always”, or by numeric ranges). Responses to the most common urinary symptoms, i.e., frequency, nocturia, urgency, sensation of incomplete emptying, hesitancy, straining to void, were consolidated into dichotomous categories corresponding with no or low severity symptomatology (lowest two points on the Likert scale), and moderate to severe symptomatology (third to fifth points on the Likert scale). A report of a moderate to severe symptom was identified as a positive response. A separate question was asked, “Do you have urinary incontinence or leakage?” to capture any incontinence outside of the one-week recall period of the LUTS Tool, and an affirmative answer was identified as a positive response.

An 11-item question regarding lower bowel symptom assessment was used to assess the frequency of bowel movements, time needed to defecate, presence of constipation, straining to defecate, pain during defecation, ability to successfully empty bowels during defecation as well as presence of chronic diarrhea and fecal incontinence. Affirmative responses to the presence of constipation or diarrhea were used to define whether the study participant had any bowel symptoms.

Participants were given an eleven-item sexual function questionnaire comprised of two sections. They were asked to complete the entire questionnaire but instructed that all questions were optional. The first section contained several general questions addressing interest in sex, partner status, if sexually active within the previous six months, and the presence of sexual dysfunction. An affirmative response to the presence of sexual dysfunction was used in the assessment of any pelvic organ dysfunction. The second section of the questionnaire addressed the characterization of sexual function, capacity to orgasm, and factors influencing sexual activity over the past 6 months.

All questionnaire responses were compared between men and women as well as between disease stages. In addition, current medications, including supplements for bowel management, were recorded.

## 3. Results

Sixty-five persons were consented and enrolled, but because of cognitive and behavioral problems, 17 persons could not complete the forms. Forty-eight subjects were able to complete all bladder, bowel, and sexual function study questionnaires (Table 1), and they form the basis of this study. All had an expanded CAG trinucleotide in the Huntingtin gene (>37) and were being followed for symptomatic HD. Participants with incomplete questionnaires were not included in this study.

### 3.1. Concomitant Pelvic Organ Symptoms

Forty-five subjects (93.8%) reported symptoms in at least one system. Nineteen subjects (39.6%) were identified with complete concomitant bladder, bowel, and sexual dysfunction, 13 (27.1%) reported impairment in two of the three pelvic organ systems queried, 13 (27.1%) reported impairment in only one organ system, and 3 subjects (6%) reported no pelvic organ symptoms.

In comparing participants with three organ system symptoms and persons with no pelvic organ symptoms, there was an inverse association with duration of disease and TFC scale, i.e., participants reporting concomitant symptoms in three pelvic organ systems were more likely to have longer duration of disease and lower functional capacity than participants reporting no pelvic organ symptoms (Table 2).

### 3.2. Urinary Symptoms

The frequency of any urinary symptom in the cohort was high, with 42 (87.5%) reporting at least one urinary symptom at sometimes, often, or always. The most frequent symptoms were incontinence (*n* = 34 (70.8%)), urinary frequency (*n* = 30 (62.5%)), urgency (*n* = 28 (58.3%)), and nocturia (*n* = 25) (52.1%)). Urinary symptom frequency was compared between men and women, (Table 3) and within each disease stage (Table 4). The frequency of any bladder symptoms was shown to increase with disease progression with an 18.2% rise in symptomatology. Although there was a clear trend of increased bladder symptoms from early to late disease, there was a mixed trend of symptom progression during mid-stage of disease.

### 3.3. Lower Bowel Symptoms

Lower bowel symptom frequency was compared between men and women (Table 3) and within each disease stage (Table 4). The presence of lower bowel symptoms was commonly reported, with 29 (60.4%) subjects reporting bowel dysfunction. Twenty-one (43.7%) reported constipation, 12 (25%) reported chronic diarrhea, with 9 (18.7%) reporting at least one to four fecal accidents per month. Eighteen (37.5%) reported straining, 12 (25%) reported difficulty or pain, 18 (37.5%) reported feeling of not emptying, and 15 (31.2%) reported unsuccessful defecation. Additionally, most subjects (83.3%) defecated at least once every one–two days, with 8 (16.7%) reporting two times or less per week and one reporting defecating once every 1–2 weeks. Twenty-three (47.9%) subjects needed less than 5 min to defecate, 25 (52.1%) needed greater than 5 min. Ten subjects (20.8%) reported using either occasional or long-term use of stool softeners, fiber supplements, or laxatives. Lower bowel symptoms were more prevalent with increasing stage of disease.

### 3.4. Sexual Activity and Function

The majority of male and female respondents were interested in sex (83.3%), had a sexual partner (68.8%), and 47.9% reported being sexually active within the past 6 months (Table 3). When asked “Do you think you have any problems with sexual function?”, 17 (35.4%) reported sexual dysfunction, 29 (60.4%) reported no dysfunction, and 2 subjects, who had not been sexually active within the past 6 months, responded unknown. However, 10 subjects (5 men and 5 women) who reported no sexual dysfunction in Section 1 of the sexual function questionnaire endorsed either impaired ability to orgasm or anorgasmia while completing the more detailed second section of the questionnaire. Thus, sexual dysfunction was reported in 27 subjects, representing 56.2% of the total respondents.

Men and women reported a similarly high degree of sexual dysfunction. Those with sexual dysfunction reported a combination of contributing factors including having a chronic illness, medications, fatigue, psychological problems, and partner incompatibility, as well as unknown causes.

The most common factor influencing sexual activity over the prior 6 months was related to self-health problems, followed by partner health problems, conflict in relationship, and lack of privacy. Sexual dysfunction was shown to increase with disease progression from early to mid-disease (Stage 1–3). Sexual activity declined severely with disease progression. Nearly all subjects (90.9%) reported being sexually active in stage 1, decreasing in stage 2 (66.7%) and stage 3 (31.2%) with no subjects reporting sexual activity in stage 4 (Table 4).

Thirty-one subjects completed Section 2 of the sexual symptom questionnaire. Overall, men reported sexual activity to be more important than women, in opposition to women who reported a higher degree of sexual enjoyment over the men. Interest in sex was high across all disease stages irrespective of sexual dysfunction or partner status.

### 3.5. Patient Vignettes

The experiences of three patients seen in our clinic emphasize the serious clinical nature of these pelvic organ problems. The first was a 40-year-old man who would urinate in his pants during his clinic visits without awareness that this was unusual and was not disturbed by his frequent fecal incontinence at home. In addition to bladder/bowel incontinence, this demonstrated the “lack of awareness” syndrome typical of HD. Second was a 50-year-old man who spent several hours each day sitting on the toilet because of bowel constipation associated with persistent obsessive thoughts about altering his diet to increase bowel function. Third was a 46-year-old woman who wanted to urinate nearly hourly during the day and then get up three times at night spending long periods urinating. This was exhausting to her caregiver and resulted in strong guilt feelings in the subject.

### 3.6. Medication Use

The majority of all subjects (38; 79%) were taking medications known to potentially impact pelvic organ function (Table 5). Of those, 36 (75%) were taking medications specifically targeted at HD symptoms. Twenty (41.7%) subjects reported taking antipsychotics, 29 (60.4%) reported taking antidepressants, 6 (12.5%) reported taking anti-seizure medication, and 2 (4.2%) reported taking anxiolytics. Bladder-specific medications were regularly used by 4 (10.4%) subjects and antidiarrheals were regularly used by 2 (4.2%). Ten (20.8%) subjects reported using either occasional, regular, or long-term use of stool softeners, fiber supplements, or laxatives. Of importance is the fact that ten subjects were on no medications and nine of them reported at least one pelvic organ system problem (four reported a problem in one organ system, two reported problems in two organ systems, and three reported problems in all three systems).

## 4. Discussion

Impairment of bladder, bowel, and sexual function has been under-recognized among individuals with HD despite significantly impacting their quality of life. In addition to motor and cognitive debility and disorders of mood, our study provides further evidence that this population experiences derangement in lower urinary tract, bowel, and sexual functioning. Approximately 40% of this cohort of persons symptomatic for HD reported concomitant bladder, bowel, and sexual dysfunction symptoms. The concurrence of all these organ dysfunctions points to the central pathophysiology of HD as a possible etiology.

### 4.1. Pathophysiology of Bladder, Bowel, and Sexual Dysfunction in HD

Although complex and incompletely characterized, the neurophysiology of bladder, bowel, and genital function are interrelated, and concomitant dysfunction of these organs often occurs due to the proximity of the central neuroanatomical pathways [12,13,17,18]. Healthy functioning of each of these systems is dependent on intact central regulation of both the autonomic and somatic nervous systems, which can be compromised in HD. Aziz et al. reported a high frequency of autonomic symptoms in patients with HD including early abdominal fullness, straining for defecation, fecal and urinary incontinence, urgency, incomplete bladder emptying, and erectile and ejaculation dysfunction [14]. Studies of autonomic nervous system function in patients in early to middle stages of HD demonstrated the existence of both sympathetic and parasympathetic derangements in persons with HD [14,15,16].

Functional imaging studies have shown neural control of bladder, bowel, and sexual function to be complex, involving many regions of the brain [22,23,24,25,26,27,28]. HD is characterized by progressive neuronal loss in the basal ganglia, and atrophy has also been reported for frontal, temporal, insular, and parietal cortical areas in HD [29,30]. This widespread neuronal loss continues throughout the disease course. Many of the regions and nuclei of the Central Nervous System (CNS) affected by HD overlap those known to mediate bladder, bowel and sexual function [31,32,33]. Not surprisingly with this pattern of atrophy, patients with HD report high rates of concomitant pelvic organ dysfunction and symptomatology.

### 4.2. Bladder Symptoms in HD

As in other degenerative CNS diseases, loss of supraspinal neurons results in a loss of inhibition of the micturition reflex, resulting in bladder overactivity with urinary urgency, frequency, nocturia, and urgency incontinence [32]. Limited studies have reported urinary symptoms in HD [2,5,6,7], and our findings corroborate them, with a high frequency of bladder dysfunction in our clinic population. In their cohort of 54 HD patients, Kolenc et al. reported a significant level of urinary symptomology in both men and women, with the most common symptoms being bladder overactivity (frequency, urgency), urinary incontinence, and symptoms of disturbed emptying [2]. Bladder symptoms increase in the general population with advancing age, in both sexes, and some of the symptomatology of this cohort could be attributed to aging and not HD alone. However, the frequency of individual lower urinary tract symptoms in this cohort, in both men and women, is nearly twice that reported in a large epidemiological study of 30,000 men and women in the US and Europe, with the mean age being approximately 7 years older than that of the current study [34].

### 4.3. Lower Bowel Symptoms in HD

Lower bowel dysfunction can result from CNS pathology such as occurs in HD. Constipation, stool retention, diarrhea, and fecal incontinence are common in persons with neurological disorders, and multiple lower bowel symptoms often coexist [35]. Impairment to the sensory perception associated with the need to defecate, diminished rectal contractility, and loss of anal sphincter motor control can result in bowel dysfunction. The frequency of lower bowel symptoms in the general population is much lower than in the current cohort when compared to gender and age-matched findings from the first National Health and Nutrition Examination Survey (NHANES) [36]. Thus, persons with HD appear to have a significantly increased rate of experiencing bowel dysfunction.

### 4.4. Sexual Symptoms in HD

Neurologic disease and conditions of chronic illness can frequently alter all phases of the sexual response, leading to impairment of sexual functioning [37]. Poeppi et al. described multiple aspects of male sexual functioning influenced by regions of the brain responsible for the regulation and triggering of both autonomic and somatosensory processing, which are known to be disrupted in HD [38]. In this cohort, subjects reported a high level of sexual dysfunction. HD, again through central autonomic dysregulation, as a likely contributing factor [39]. Sexual function can be influenced by psychological (arousal, desire, emotions) or physiological factors (genital responses, hormonal influence, medications, alcohol), or a combination of both. Fedoroff et al. reported on sexual disorders in HD, suggesting that both motor and cognitive aspects of HD including difficulty with maintaining attention and concentration may interfere in the ability to achieve orgasm [40].

More recently, the availability of neuroimaging through the use of Single Photon Emission Computed Tomography (SPECT), Positron Emission Tomography (PET), Functional Magnetic Resonance Imaging (fMRI), Blood Oxygen Level-Dependent Functional Magnetic Resonance Imaging BOLD fMRI, and Magnetoencephalography (MEG) studies have shown that sexual arousal stimuli provokes activity in a broad neural network of cortical and subcortical brain areas that are known to be associated with emotional, cognitive, motivational, and physiological processing, all of which are commonly impaired in persons with HD [26,27,28].

### 4.5. Impact of HD-Related Medications on Bladder, Bowel, and Sexual Symptoms

In our cohort, 38 (79%) subjects were being treated with one or more centrally acting medications to treat chorea, mood instability, obsessive compulsive disorder, and depression. Antipsychotics, antidepressants, anti-seizure medication, and benzodiazepines comprise the mainstay of pharmacological treatment options for patients with HD, either in monotherapy or in combination. These medications are centrally acting and thus may have an impact on bladder, bowel, and sexual function. Use of these drugs may impair bladder function, which can present in the form of either urinary retention or incontinence [41]. Similarly, gastrointestinal hypomotility is a reported side effect of some antipsychotic medication use, typically constipation [42]. Sexual dysfunction can also be a side effect of psychotropic drugs. These drugs are usually inhibitory in nature and may negatively affect all phases of the sexual response cycle (desire, arousal, and orgasm) [43]. As a result of their common use, it is very difficult to tease apart the contributions of medications used in HD to the observed bowel, bladder, and sexual dysfunction.

However, it is unlikely that medication is the only factor driving pelvic organ dysfunction in this study because of the high rate (39.6%) of concomitant symptoms in all pelvic organ systems in the entire group and the frequent occurrence of symptoms in nine of the 10 subjects taking no drugs at all. In addition, some of the psychiatric features often seen in HD, such as perseveration, obsessive compulsive thoughts and behaviors, and lack of awareness, can further impact pelvic organ symptoms and management.

Confounding variables, such as cognitive and memory impairment were also shown to impact the ability for some subjects to report accurately. Studies have shown that 30–50% of persons with HD often underestimate self-reports of disease manifestations, progression of disease, and impact of impairment [44]. This phenomenon may have led to underreporting of bladder, bowel, and sexual symptoms in the current study. Some subjects reporting on sexual function may not have recognized difficulty achieving orgasm or anorgasmia as a symptom of sexual dysfunction. Those receiving assistance with completing their questionnaires may have experienced embarrassment in discussing the questions specifically regarding sexual dysfunction, leading to decreased reporting of dysfunction.

In conclusion, there is a high frequency of concomitant bladder, bowel, and sexual symptoms in persons with HD. The pervasiveness of this symptomatology suggests that HD pathophysiology is a major contributing factor. Given this frequency, it is important for health care providers to recognize these symptoms and account for them in the overall treatment and management of patients with HD as they would other functional deficits. Many patients will present with a combination of urinary, bowel, and sexual dysfunction, which should be addressed in a holistic manner.

## Figures and Tables

**Table 1 jpm-11-00714-t001:** Participant demographics.

Variable(s)	
Age	
Range (mean)	25–78 (48.9)
CAG repeats	
Range (mean)	38–61 (43.8)
TFC scale rating	
Range (mean)	1–13 (6.5)
Disease stage	
Range (mean)	1–4 (2.5)
Age of Disease Onset	
Range (mean)	16–71 (42.0)
Duration of Disease	
Range (mean)	1–22 (6.8)
Gender, *n* (%)	
Male	26 (54.2)
Female	22 (45.8)
Race, *n* (%)	
White	45 (93.8)
Black	1 (2.0)
American Indian/Alaska Native	2 (4.2)
Ethnicity, *n* (%)	
Hispanic	0 (0.0)
Not Hispanic	48 (100.0)
Relationship status, *n* (%)	
Married	26 (54.2)
Living with partner	4 (8.3)
Single	18 (37.5)

**Table 2 jpm-11-00714-t002:** Organ systems and concomitant dysfunction.

Symptomatic Organ System(s) Dysfunction	None *n* = 3 (6.3%)	One *n* = 13 * (27.1%)	Two *n* = 13 * (27.1%)	Three *n* = 19 * (39.6%)
Age				
Range (mean)	32–67 (44.3)	29–78 (48.9)	28–66 (43.3)	25–77 (51.7)
CAG				
Range (mean)	40–42 (40.7)	39–49 (43.5)	41–61 (46)	39–53 (43)
TFC				
Range (mean)	5–13 (10.3)	2–13 (7.9)	1–13 (6)	1–13 (5.9)
Disease Stage				
Range (mean)	1–3 (1.7)	1–4 (2.2)	1–4 (2.9)	1–4 (2.5)
Age of Onset				
Range (mean)	30–52 (38.3)	24–67 (44.4)	16–60 (37.7)	22–71 (43)
Duration of Disease				
Range (mean)	1–15 (5.7)	1–16 (6.5)	1–12 (5.5)	1–22 (8.1)
Gender, *n* (%)				
Male	2	8	7	10
Female	1	5	6	9

* In total, 45 subjects 45 (93.8%) subjects reported impairment in at least one of the three pelvic organ systems.

**Table 3 jpm-11-00714-t003:** Bladder, lower bowel, and sexual symptoms stratified by gender.

Symptom Measure(s)	All *n* = 48 (%)	Males *n* = 26 (%)	Females *n* = 22 (%)
**Bladder Symptom(s)**			
Any urinary symptom			
No (i.e., never/rarely)	6 (12.5)	4 (15.4)	2 (9.0)
Yes (i.e., sometimes/often/always)	42 (87.5)	22 (84.6)	20 (91)
Frequency			
Never/rarely	18 (37.5)	11 (42.3)	7 (31.8)
Sometimes/often/always	30 (62.5)	15 (57.7)	15 (68.2)
Voids			
1–7 per day	27 (56.3)	15 (57.7)	12 (54.5)
8–14+ per day	21 (43.7)	11 (42.3)	10 (45.5)
Nocturia			
1–2 per night	23 (47.9)	10 (38.5)	13 (59.1)
3–4+ per night	25 (52.1)	16 (61.5)	9 (40.9)
Urgency			
Never/rarely	20 (41.7)	13 (50)	7 (31.8)
Sometimes/often/always	28 (58.3)	13 (50)	15 (68.2)
Incomplete empty			
Never/rarely	31 (64.6)	17 (65.4)	14 (63.6)
Sometimes/often/always	17 (35.4)	9 (34.6)	8 (36.4)
Hesitancy			
Never/rarely	31 (64.6)	14 (53.8)	17 (77.3)
Sometimes/often/always	17 (37.5)	12 (46.2)	5 (22.7)
Strain to urinate			
Never/rarely	38 (79.2)	18 (69.2)	20 (90.9)
Sometimes/often/always	10 (20.8)	8 (30.8)	2 (9.1)
Urinary incontinence/leakage			
No	14 (29.2)	9 (34.6)	5 (22.7)
Yes	34 (70.8)	17 (65.4)	17 (77.3)
**Lower bowel symptom(s)**			
Any lower bowel symptom(s)			
No	19 (39.6)	12 (46.2)	7 (31.8)
Yes *	29 (60.4)	14 (53.8)	15 (68.2)
Constipation			
No	27 (56.3)	16 (61.5)	11 (50.0)
Yes	21 (43.7)	10 (38.5)	11 (50.0)
Chronic diarrhea			
No	36 (75.0)	20 (76.9)	16 (72.7)
Yes	12 (25.0)	6 (23.1)	6 (27.3)
BM incontinence/accident			
No	39 (81.3)	20 (76.9)	19 (86.4)
Yes	9 (18.7)	6 (23.1)	3 (13.6)
BM frequency			
1–2 times per 1–2 days	40 (83.3)	23 (88.5)	17 (77.3)
≤2 per week	8 (16.7)	3 (11.5)	5 (22.7)
BM time			
<5 min	23 (47.9)	13 (50.0)	10 (45.5)
≥5 min	25 (52.1)	13 (50.0)	12 (54.5)
Strain while having BM			
Never/rarely	30 (62.5)	17 (65.4)	13 (59.1)
Sometimes/often/always	18 (37.5)	9 (34.6)	9 (40.9)
Difficult or painful BM			
Never/rarely	36 (75.0)	20 (76.9)	16 (72.7)
Sometimes/often/always	12 (25.0)	6 (23.1)	6 (27.3)
Feeling of not emptying after BM			
Never/rarely	30 (62.5)	15 (57.7)	15 (68.2)
Sometimes/often/always	18 (37.5)	11 (42.3)	7 (31.8)
Unsuccessful having BM			
Never/rarely	33 (68.8)	20 (76.9)	13 (59.1)
Sometimes/often/always	15 (31.2)	6 (23.1)	9 (40.9)
Sexual activity and function			
**Section 1**			
Interest in sex			
Yes	40 (83.3)	20 (76.9)	20 (90.9)
No	8 (16.7)	6 (23.1)	2 (9.1)
Sexual partner			
Yes	33 (68.8)	18 (69.2)	15 (68.2)
No	15 (31.2)	8 (30.8)	7 (31.8)
Sexually active in last 6 months			
Yes	23 (47.9)	9 (34.6)	14 (63.6)
No	25 (52.1)	17 (65.4)	8 (36.4)
Problems with sexual function			
Yes	27 (56.2)	15 (57.7)	12 (54.5)
No	19 (39.6)	10 (38.5)	9 (40.9)
Unknown	2 (4.2)	1 (3.8)	1 (4.6)
**Section 2** **^†^**	**All** ***n*** **= 31(%)**	**Males** ***n*** **= 16(%)**	**Females** ***n*** **= 15(%)**
Importance of sexual activity			
Very important	15 (48.4)	10 (62.5)	5 (33.3)
Important or somewhat important	8 (25.8)	3 (18.8)	5 (33.3)
Somewhat unimportant or not important at all	8 (25.8)	3 (18.8)	5 (33.3)
Enjoyment of sexual activity			
Never/rarely enjoy	9 (29.0)	7 (43.8)	2 (13.3)
Sometimes enjoy	10 (33.3)	4 (25.0)	6 (40.0)
Fully enjoy	12 (38.7)	5 (31.2)	7 (46.7)

* Any bowel symptom(s) defined as the presence of constipation and/or chronic diarrhea. **^†^** Sexual function measures–Section 2 was not completed by all participants.

**Table 4 jpm-11-00714-t004:** Bladder, lower bowel, and sexual symptoms stratified by disease stage.

Symptom Measure(s)	Stage 1 * *n* = 11 (%)	Stage 2 * *n* = 12 (%)	Stage 3 * *n* = 16 (%)	Stage 4 * *n* = 9 (%)
**Bladder Symptom(s)**				
Any urinary symptom				
No (i.e., never/rarely)	2 (18.2)	0 (0.0)	4 (18.8)	0 (0.0)
Yes (i.e., sometimes/often/always)	9 (81.8)	12 (100)	12 (75)	9 (100)
Frequency				
Never/rarely	6 (54.5)	4 (33.3)	6 (37.5)	2 (22.2)
Sometimes/often/always	5 (45.5)	8 (66.7)	10 (62.5)	7 (77.8)
Voids				
1–7 per day	7 (63.6)	6 (50.0)	9 (56.2)	4 (44.4)
8–14+ per day	4 (36.4)	6 (50.0)	7 (43.8)	5 (55.6)
Nocturia				
1–2 per night	7 (63.6)	7 (58.3)	7 (43.8)	2 (22.2)
3–4+ per night	4 (36.4)	5 (41.7)	9 (56.2)	7 (77.8)
Urgency				
Never/rarely	5 (45.5)	5 (41.7)	7 (43.8)	3 (33.3)
Sometimes/often/always	6 (54.5)	7 (58.3)	9 (56.2)	6 (66.7)
Incomplete empty				
Never/rarely	5 (45.5)	6 (50.0)	13 (81.3)	7 (77.8)
Sometimes/often/always	6 (54.5)	6 (50.0)	3 (18.8)	2 (22.2)
Hesitancy				
Never/rarely	9 (81.8)	7 (58.3)	10 (62.5)	5 (55.6)
Sometimes/often/always	2 (18.2)	5 (41.7)	6 (37.5)	4 (44.4)
Strain to urinate				
Never/rarely	10 (90.9)	9 (75.0)	12 (75.0)	7 (77.8)
Sometimes/often/always	1 (9.1)	3 (25.0)	4 (25.0)	2 (22.2)
Urinary incontinence/leakage				
No	4 (36.4)	3 (25.0)	6 (37.5)	1 (11.1)
Yes	7 (63.6)	9 (75.0)	10 (62.5)	8 (88.9)
Lower bowel symptom(s)				
No	6 (54.5)	6 (50.0)	6 (37.5)	1 (11.1)
Yes *	5 (45.5)	6 (50.0)	10 (62.5)	8 (88.9)
Constipation				
No	7 (63.6)	7 (58.3)	10 (62.5)	3 (33.3)
Yes	4 (36.4)	5 (41.7)	6 (37.5)	6 (66.7)
Chronic diarrhea				
No	9 (81.8)	10 (83.3)	11 (68.8)	6 (66.7)
Yes	2 (18.2)	2 (16.7)	5 (31.2)	3 (33.3)
BM incontinence/accident				
No	10 (90.9)	12 (100.0)	10 (62.5)	6 (66.7)
Yes	1 (9.1)	0 (0.0)	6 (37.5)	3 (33.3)
BM frequency				
1–2 times per 1–2 days	10 (90.9)	9 (75.0)	15 (93.7)	6 (66.7)
≤2 per week	1 (9.1)	3 (25.0)	1 (6.3)	3 (33.3)
BM time				
<5 min	5 (45.5)	7 (58.3)	8 (50.0)	3 (33.3)
≥5 min	6 (54.5)	5 (41.7)	8 (50.0)	6 (66.7)
Strain while having BM				
Never/rarely	7 (63.6)	6 (50.0)	10 (62.5)	7 (77.8)
Sometimes/often/always	4 (36.4)	6 (50.0)	6 (37.5)	2 (22.2)
Difficult or painful BM				
Never/rarely	8 (72.7)	9 (75.0)	11 (68.8)	7 (77.8)
Sometimes/often/always	3 (27.3)	3 (25.0)	5 (31.2)	2 (22.2)
Feeling of not emptying after BM				
Never/rarely	6 (54.5)	7 (58.3)	10 (62.5)	6 (66.7)
Sometimes/often/always	5 (45.5)	5 (41.7)	6 (37.5)	3 (33.3)
Unsuccessful having BM				
Never/rarely	8 (72.7)	9 (75.0)	10 (62.5)	6 (66.7)
Sometimes/often/always	3 (27.3)	3 (25.0)	6 (37.5)	3 (33.3)
**Sexual Activity and Function**				
**Section 1**				
Interest in sex				
Yes	11 (100.0)	12 (100.0)	11 (68.8)	6 (66.7)
No	0 (0.0)	0 (0.0)	5 (31.2)	3 (33.3)
Sexual partner				
Yes	10 (90.9)	9 (75.0)	8 (50.0)	6 (66.7)
No	1 (9.1)	3 (25.0)	8 (50.0)	3 (33.3)
Sexually active in last 6 months				
Yes	10 (90.9)	8 (66.7)	5 (31.2)	0 (0.0)
No	1 (9.1)	4 (33.3)	11 (68.8)	9 (100.0)
Problems with sexual function				
Yes	5 (45.5)	8 (66.7)	10 (62.5)	4 (44.4)
No	6 (54.5)	4 (33.3)	6 (37.5)	3 (33.3)
Unknown	0 (0.0)	0 (0.0)	0 (0.0)	2 (22.2)
**Section 2** ***n* = 31 ^†^**	***n* = 10 (%)**	***n* = 8 (%)**	***n* = 9 (%)**	***n* = 4 (%)**
Importance of sexual activity				
Very important	4 (40.0)	6 (75.0)	3 (33.3)	2 (50)
Important or somewhat important	2 (20.0)	1 (12.5)	5 (55.6)	0 (0.0)
Somewhat unimportant or not important at all	4 (40.0)	1 (12.5)	1 (11.1)	2 (50)
Enjoyment of sexual activity				
Never/rarely enjoy	0 (0.0)	2 (25.0)	3 (33.3)	4 (100)
Sometimes enjoy	5 (50.0)	2 (25.0)	3 (33.3)	0 (0.0)
Fully enjoy	5 (50.0)	4 (50.0)	3 (33.3)	0 (0.0)

* Stage 1, 2, 3, and 4 corresponds to TFC scale ratings 11–13, 7–10, 3–6, and 1–2, respectively. **^†^** Sexual function measures–Section 2 was not completed by all participants.

**Table 5 jpm-11-00714-t005:** Medication classes and possible effects on pelvic organ function.

Medication Class	Possible Effect(s)
Antipsychotics (e.g., fluphenazine, olanzapine, haloperidol, quetiapine, aripiprazole)	Depressed bladder contractility, urinary retention, urinary incontinence, constipation
Antidepressants: Selective Serotonin Reuptake Inhibitors (SSRI’s) and Serotonin-Norepinephrine Reuptake Inhibitors (SNRIs) (e.g., citalopram, escitalopram, sertraline, venlafaxine, duloxetine)	Overactive bladder, urinary retention, constipation, diarrhea, diminished sexual desire, erectile function, delayed or absent orgasm
Antiepileptics (e.g., divalproex sodium, lamotrigine, carbamazepine)	Constipation, diarrhea
Benzodiazepines (e.g., clonazepam, lorazepam)	Depressed bladder contractility, constipation, diarrhea

## Data Availability

All data available upon request from corresponding author.

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
