# Peer review of "High Frequency of Concomitant Bladder, Bowel, and Sexual Symptoms in Huntington’s Disease: A Self-Reported Questionnaire Study"

_jpm, 2021, doi:10.3390/jpm11080714_

Round 1

Reviewer 1 Report

The authors built this interesting study based on the clinical experience they gained on Huntington's disease (HD). Over time, among patients with this neurodegenerative disease they have noted adverse patient reports of symptoms such as urinary frequency, urgency, incontinence (both urinary and fecal), constipation, chronic diarrhea and sexual dysfunction, often in combination in the same person. The authors consider that current texts of HD rarely pay attention to the association of the triad of motor, cognitive and psychological symptoms and the concomitant pelvic organ dysfunctions. This is contrast to the classical recognition of bladder, bowel and sexual symptoms and signs in many other neurological diseases.

Title                                                                                                                    I suggest replacing the term prevalence with the term frequency or other. It is more correct and adherent to the type of study.  Prevalence is a measure of frequency, a rate for epidemiological use borrowed from statistics. Prevalence is the ratio between the number of health events detected in a population at a given time and the number of individuals in the population observed in the same period.

Methods                                                                                                            . the description of the inclusion and selection of cases is complete and well structured.                                                                                                          . The use of TFC scale to measure disease stage is correct                                 . The detailed description of the questions included in the questionnaire regarding urinary symptoms, lower bowel symptoms and sexual dysfunctions is detailed and comprehensive.

 Results                                                                                                              The results are described in detail and the tables showing the data and the percentages calculated for each item are clear and easy to read. All calculations are correct. I can propose, in order to facilitate the reading of tables 3 and 4, to place paragraphs 3.3 (Lower bowel symptoms) and 3.4 (Sexual activity and function) which can be found on pages 11 and 12 (from line 147 to line 184) in the publication of the work before table 3 (on page 5), immediately after paragraph 3.2 Urinary symptoms. Probably moving the paragraphs would help the reader by anticipating what numbers and percentages of each pelvic organ dysfunction report in the all-encompassing table.                                                                                                               Having reported experiences of three patients is interesting and emphasizes the serious nature of the pelvic organ problems in HD.                             Table 5 is important, because it must remind us of the impact of HD related medications on bladder, bowel and sexual clinical probems, as it is amply reiterated in the subsequent discussion.

Discussion                                                                                                    The authors analyze the data collected and discuss them by comparing them with what is reported in the literature, initially proposing an evaluation of the pathophysiology of the dysfunctions studied through scientific and clinical bases, including recently published ones. The remaining discussion on the individual symptoms studied, and on the therapies, is important and very well structured. The reference to confounding variables, such as cognitive and memory impairment, is also important, since it may also have led to some underestimation of the symptoms.

References are complete and updated.

In conclusion, the paper is certainly worthy of publication. The authors may consider the suggestions to move the paragraphs of the results and to modify title as I have suggested, however these are not significant problems.

Author Response

Response to Reviewer:

We thank the reviewer for the useful comment.  As suggested we have changed the word “prevalence” to “frequency” in the Title and throughout the text.  We have also checked to be certain that the paragraphs and the tables are in the correct order.  These are all the changes suggested by the reviewer. 

Reviewer 2 Report

  • This is a very interesting manuscript highlithing the hypothesis that HD is systemic more than simply a nervous system illness.  Background introduction might be improved with updated references regarding instrumental evidence of autonomic dysfunction in HD such as the following:
      • Autonomic disorders and myocardial 123I-metaiodobenzylguanidine scintigraphy in Huntington's disease. Assante R, Salvatore E, Nappi C, Peluso S, De Simini G, Di Maio L, Palmieri GR, Ferrara IP, Roca A, De Michele G, Cuocolo A, Pappatà S, De Rosa A. J Nucl Cardiol. 2020 Aug 16. doi: 10.1007/s12350-020-02299-7.
      • Anorectal Dysfunction in Presymptomatic Mutation Carriers and Patients with Huntington's Disease. Jan Kobal, Kolenc Matej, Matic Koželj, Simon Podnar. J Huntingtons Dis. 2018;7(3):259-267. doi: 10.3233/JHD-170280.
      • Non-motor symptoms in Huntington's disease: a comparative study with Parkinson's disease. Aldaz T, Nigro P, Sánchez-Gómez A, Painous C, Planellas L, Santacruz P, Cámara A, Compta Y, Valldeoriola F, Martí MJ, Muñoz E. J Neurol. 2019 Jun;266(6):1340-1350. doi: 10.1007/s00415-019-09263-7. Epub 2019 Mar 5.
      • Early enteric neuron dysfunction in mouse and human Huntington disease. Sciacca S, Favellato M, Madonna M, Metro D, Marano M, Squitieri F. Parkinsonism Relat Disord. 2017 Jan;34:73-74. doi: 10.1016/j.parkreldis.2016.10.017. Epub 2016 Nov 1.
    • The cohort's size, even though relatively small, highlights the hypothesis that HD is a systemic illness involving several organs. I think the authors should anyway list the relatively small size of the sample as a limitation in Discussion.   
    • It should be underlined that the evidence is preliminary (requiring larger-scale confirmation) also in the title: p.e. Self-reported Questionnaire of Bladder, Bowel and Sexual Symptoms in Huntington’s Disease: a preliminary study. 
    • The study uses validated self-questionnaires, but some patients in latter stages were assisted in completing them determining a methodologic issue. I would suggest to include this as a possible bias and limitation in Discussion.
    • The comparison between symptom prevalence in the sample with previous epidemiological studies in the general population might have been quantitative, besides qualitative. May the authors report the data present in literature regarding symptom prevalence in general population in the discussion?
    • The results are clearly expressed but the data presentation tables seem excessively long and dispersive, authors could consider to simplify the tables and/or to merge some of the tables together (i.e. table 3 and 4) or put some of the table material into an appendix. 

Author Response

(The authors gave the same response as above.)
